# Increasing Doses of Bacterial Phytase (*Citrobacter braakii*) Improves Performance and Carcass Characteristics of Pigs in Growing and Finishing Phases

**DOI:** 10.3390/ani12192552

**Published:** 2022-09-24

**Authors:** Caio Abércio da Silva, Marco Aurélio Callegari, Cleandro Pazinato Dias, Kelly Lais de Souza, Rafael Humberto de Carvalho, Leandro Alebrante, Claudia Cassimira da Silva Martins, Augusto Heck, Vitor Barbosa Fascina

**Affiliations:** 1Department of Zootechnology, Center of Agrarian Sciences, State University of Londrina, Londrina 86057970, Brazil; 2Akei Animal Research, Fartura 18870970, Brazil; 3DSM Nutritional Products Ltda., Jaguaré 05321900, Brazil

**Keywords:** calcium, enzyme, phytate, phytic acid, swine

## Abstract

**Simple Summary:**

Phytases of bacterial origin are the exogenous enzymes most widely used in the diets of monogastric animals, acting on the hydrolysis of phytate (myo-inositol 1,2,3,4,5,6-hexakis [dihydrogen] phosphate) to release the phosphate from this complex. However, the efficiency of phytase varies according to the aspects inherent to its own structure and origin, the age and physiological state of the animal, and the dietary composition. Overall, we found that supplementation with phytase between 1500 and 3000 units of phytase (FYT)/kg of feed in diets based on corn and soybean meal with a reduction in inorganic phosphorus (0.13%) and calcium (0.11%) improves the daily weight gain (DWG) and live weight (LW) of animals in the growing phase, which affects performance on finishing phases and slaughter body weight. This input could result in a reduction in the use of phosphorus sources, improving environmental issues with an increase in the main zootechnical parameters.

**Abstract:**

The effects of increasing doses of bacterial phytase (*Citrobacter braakii*) on performance and carcass characteristics of growing-finishing pigs was evaluated. A total of 120 barrows weighing 25.16 ± 2.80 kg and 68 days old were submitted to five treatments: positive control diet (PC) containing inorganic phosphorus (P) and limestone (Ca); negative control (NC) with reductions in P (by 0.13%) and Ca (by 0.11%); and three NC diets supplemented with 1500 (NC15), 3000 (NC30) and 4500 (NC45) units of phytase (FYT)/kg. The daily weight gain (DWG) in growth phase I (68–91 days) was higher in the PC, NC15 and NC30, compared to the NC (1.06, 1.06, 1.06 vs. 0.95, respectively). The DWG in finishing phase II (141–156 days) was higher in the NC15 (1.20 kg) and NC30 (1.14 kg) than in the NC45 (0.94 kg). The final weights (LW156) in the NC15 (122.95 kg LW) were higher than NC (116.47 kg LW) and NC45 (114.43 kg LW). Over the entire period, a quadratic effect (2012 FYT) was observed for the DWG. The increasing levels of phytase in corn and soybean meal-based diets improved the DWG and carcass traits; however, the addition of more than 3000 FYT/kg of feed should be carefully studied to determine enzyme viability.

## 1. Introduction

Phytate or phytic acid, present in foods of plant origin, is considered an antinutritional component because it has an electronegative charge in the intestinal lumen, favoring the formation of insoluble chelates with various cations, such as calcium (Ca), zinc (Zn) and copper (Cu), and decreasing the availability of these dietary minerals. The molecule acts by not only binding to dietary Ca, such as calcium carbonate [1,2], but also interacting with the intrinsic Ca contained in ingredients, such as corn and soybean meal, thus reducing their availability in the diet [3,4].

Additionally, dietary phytate has adverse effects on the digestibility of other nutrients, compromising the performance of monogastric animals, as reported by Woyengo et al. [5], who found that corn-based diets containing 2% phytate reduced the daily weight gain (DWG) and worsened the feed conversion of piglets in the nursery phase.

Phytase is an enzyme that hydrolyses phytate (or phytic acid) and, consequently, increases the availability of phosphorus (P), other nutrients and energy bound to phytate in food. Especially related to energy, this enzyme shows a similar result compared with the exogenous carbohydrases, which improve the availability of food energy and together can potentiate this effect [6]. Commercially, there are several sources of exogenous phytase available for use in monogastric nutrition, the synthesis of which is derived from isolated yeast or bacteria [7,8,9], such as *Buttiauxella* spp. [10,11,12], *Aspergillus niger* [13,14], *Citrobacter braakii* [8,15,16] and *Escherichia coli* [16,17,18,19,20,21]. These enzymes are classified into two groups, 3-phytase or 6-phytase, according to the site phytate on the molecule where the orthophosphate is hydrolyzed [22,23].

The efficiency of phytase varies according to the aspects inherent to its own structure and origin, the age and physiological state of the animal, the dietary composition, and the exposure condition/arrangement of the phytate molecule in the food, among other factors [11,24]. Despite these sources of variation, the effects of this enzyme, which is used in several phases, but more concentrated in pigs in the growth and finishing phases, are evident [2,25]. The common doses are on the order of 500 units of phytase (FYT)/kg [12,26]; at these values, less than 50% of the dietary phytate is commonly hydrolyzed.

Studies with higher doses of phytase, a concept called "overdosing", represented by the use of more than 1500 FYT/kg of feed [8,27], have shown results in the release of more P and other nutrients, with responses of increased performance [8,28] and positive repercussions on carcass parameters [8,29]. However, to optimize the implementation of this concept, it is essential to know the source and concentration of dietary phytate, the distribution of myo-inositol after hydrolysis, the amino acid levels and ion balance in the diet and the genetics and age of the animal [27].

The objective of the present study was to evaluate the effects of “overdosing” microbial phytase (*C. braakii*) within a wider range of doses on the performance and carcass characteristics of pigs in the growth and finishing phases using corn and soybean meal-based diets.

## 2. Materials and Methods

A total of 120 castrated pigs (Camborough × PIC^®^ 337), with a weight of 25.16 ± 2.80 kg and an age of 68 days, were used. The animals were divided into a randomized block design (according to their initial weight), with five treatments and eight replicates, with three pigs per pen which was considered the experimental unit. The air temperature and relative humidity during the experimental period were 24.5 ± 4.9 °C and 64.8 ± 12.7%.

The animals were subjected to 5 treatments, based on the dietary levels of Ca and P and the inclusion of varying levels of phytase: T1, positive control (PC), a diet with inorganic P and Ca provided through dicalcium phosphate and limestone, respectively; T2, negative control (NC), a diet with a reduction in the amounts of inorganic phosphorus (av. P) (−0.13%) and total Ca (−0.11%); T3, NC15, a diet containing 1500 units of phytase (FYT)/kg of feed; T4, NC30, a diet containing 3000 FYT/kg of feed; and T5, NC45, and a diet containing 4500 FYT/kg of feed.

All the diets with the addition of phytase led to a reduction in the availability of inorganic phosphorus (−0.13%) and calcium (−0.11%). RONOZYME^®^ HiPhos (DSM Nutritional Products, Brazil) was used as a phytase source, with 75, 150 and 225 g/ton of feed in treatments T3, T4 and T5, respectively.

The animals were subjected to a feeding program with four phases: growth I (68–91 days old); growth II (92–112 days old); finish I (113–140 days old); and finish II (141–156 days old) (Table 1). The diets based on corn and soybean meal were formulated to meet the minimum recommendations described by Rostagno et al. [30]. Diets and water were provided *ad libitum* throughout the experimental period that lasted 88 days (68 to 156 days old).

Daily weight gain (DWG), daily feed intake (DFI), and feed conversion ratio (FCR) were evaluated according to the experimental phases and for the entire study period. At 156 days old, the animals were sent for slaughter; they were subjected to a 12-hour fast before transport, with no limitations in water supply, which remained at will until the time of slaughter. According to the instructions for the humane slaughter of pigs, the animals were desensitized by means of a device (Petrovina IS 2000) with two electrodes (350 V and 1.3 A) for approximately 3 s and then bled through the large vessels of the neck.

After bleeding, the carcasses were scalded and eviscerated and followed the processing line, where the longitudinal division of the carcasses into two similar parts was performed. Subsequently, the half-carcasses were weighed and stored at 2 ± 1 °C for 24 h in a cooling chamber. Prior to storage, each left half of the carcass was subjected to electronic evaluation (Hennessy Grade Probe, Hennessy Grading Systems, Auckland, NZ). The variables obtained were the carcass weight (kg); carcass yield (%); backfat thickness (mm) measured at point P2, that is, 59 mm lateral to the dorsal midline of the carcass immediately adjacent to the last caudal rib on one side of the carcass; depth of the longissimus dorsi muscle (mm), also measured at point P2; percentage of lean meat in the carcass (%); and lean meat yield from the carcass (kg). Lean meat yield from the carcass was obtained by multiplying the carcass weight by the percentage of lean meat.

The lean meat yield was also calculated according to the European Carcass Classification, where the letters S, E, U, R, O and P represent the following percentages of lean meat in the carcass: >60, 55–60, 50–55, 45–50, 40–45 and <40%, respectively.

The pens were used as the experimental unit for evaluating the performance parameters of the carcass. The data were subjected to analysis of variance (ANOVA), and the means were compared by Tukey’s test. Nonparametric data were analyzed using the Chi-square test. In both tests, the significance level for determining differences between the means was 0.05. In addition, all the performance and carcass parameters were subjected to regression analysis at 0 (negative control), 1500, 3000 and 4500 FYT/kg of feed. R statistical software version 3.5.0 was used for all statistical analyses.

## 3. Results

In growing phase I (68–91 days), a quadratic effect (*p* < 0.05; Table 2) was observed for FCR, with the best level of phytase estimated at 2,575 FYT/kg of feed; For DWG, within growing phase I, phytase had a positive effect between treatments (*p* < 0.05), with the values being 105 (+10.9%), 104 (+10.8%), 105 (+10.9%) and 21 g (+2.1%) higher in the PC (positive control), NC15 (1500 FYT/kg of feed), NC30 (3000 FYT/kg of feed) and NC45 (4500 FYT/kg of feed) treatments, respectively, compared to the NC treatment (negative control). The DFI and live weight at 68 days (LW68) which marked the end of growing phase I, showed no differences between the treatments analyzed (*p* > 0.05) or regression effect.

As a result of the relatively high DWG in the growing phase I, the growing phase II: 92–112 days) also showed heavier animals at 92 days (LW92) for the PC, NC15, NC30 and NC45 treatments, with respective increases in body weight of 2.83 (+6.0%), 2.62 (+5.6%), 2.42 (5.2%) and 0.69 kg (+1.5%) relative to animals in the NC treatment (*p* < 0.05). There were no differences between treatments (*p* > 0.05) for DWG, DFI and FCR

At 112 days old, when pigs reached the finishing phase I, the best results were observed in the animals from the PC, NC15 and NC30 treatments (*p* < 0.05). Increases in body weight of 3.30 (+4.76), 4.45 (+6.4%) and 3.30 (+4.7%) kg were observed in the PC, NC15 and NC30 treatments, respectively, compared with the NC treatment. There was no difference between treatments, nor regression effect for DWG, DFI or FCR in this phase.

In the finishing II (141–156 days), the parameters DWG, DFI and FCR showed a quadratic regression effect (*p* < 0.05), with the best doses of phytase being estimated at 1466, 1740, and 1225 FYT, respectively. There was a difference in the DWG and LW156 parameters (*p* < 0.05), with the DWG being higher in the NC15 (1.20 kg, +27.0%) and NC30 (1.14 kg, +20.7%) treatments than in the NC45 (0.94 kg) treatment; the PC treatment (1.06 kg; +12.6%) presented an intermediate result for this parameter. The animals that received 1500 FYT (i.e., those in the NC15 group) had higher final weights (LW156) than the animals in the NC (6.48 kg, +5.6%) and NC45 (8.52 kg, +7.4%) treatments, respectively (*p* < 0.05).

The experimental results, considering the entire evaluation period (68–156 days), showed a quadratic regression effect for the parameter DWG (*p* < 0.05), which increased up to the estimated level of 2012 FYT. Similarly, the DWGs of animals in the NC15 and NC30 treatments were 6.6% and 4.7% higher, respectively, than that of the animals in the NC treatment (1.05 kg; *p* < 0.05).

In the evaluation of carcass characteristics (Table 3), a quadratic regression effect (*p* < 0.05) was observed for live slaughter weight (kg) and carcass weight (kg), with the best phytase levels estimated at 2146 and 2101 FYT, respectively. Both parameters showed a difference between the groups analyzed (*p* < 0.05), where animals in the NC15 group had increases of 6.44 (+5.6%) and 4.86 kg (+5.9%) in live slaughter weight and carcass weight, respectively, compared to those in the NC45 treatment (*p* < 0.05). The addition of phytase to the NC15 and NC30 treatments increased the weights of the carcasses by 4.1 (+4.9%) and 2.6 kg (+3.2%), respectively, compared to the carcasses of animals in the NC treatment. The other parameters (Table 3) did not differ between treatments (*p* < 0.05).

According to the European standard for carcass classification (Table 4), the carcasses evaluated were mostly classified in the U (50–55% lean meat; *n* = 55) and E (55–60% lean meat; *n* = 29) categories. However, a greater number of animals with carcasses in category E were from the NC45 treatment (*n* = 11) than the other treatments analyzed (*p* > 0.05). Two carcasses in the NC group were less than 50% lean. 

## 4. Discussion

All the feeds that contained phytase had reduced levels of Ca and P (Table 1); however, the zootechnical performances of the NC15 and NC30 treatments, especially for the parameters DWG and liveweight (LW), in the different phases were improved compared with that of the PC treatment (Table 2). This condition represents a better nutritional use of the diet, as observed in other studies in which an overdose of phytase was administered [8,10], allowing us to assume that such positive effects also extend to environmental and economic issues.

The DWGs of the animals in the NC group were lower (*p* < 0.05) only in growth phase I than those of the animals in the other treatments (PC, NC15 and NC30), a result similar to that obtained by Silva et al. [8], indicating that nutrient demands are higher in the early stages of fattening. This finding is also reflected in the higher DWG of the animals in growth phase I; the live weight at 91 days (LW91), at the end of this phase, was higher for the PC, NC15 and NC30 treatments than for the NC treatment (Table 2; *p* < 0.05), with subsequent repercussions in the weights at 112 (LW112) and 156 (LW156) days. In particular, the treatment with 1500 FYT/kg of feed showed the best and longest-lasting responses in the DWG and LW parameters.

Better results in the DWG and LW of growing pigs supplemented with phytase were reported by Dersjant-Li et al. [12]. The authors, working with 250, 500 and 1000 FYT/kg of feed in diets with reduced levels of Ca and P (NC), found improvements in DWG of 3.5, 7.2 and 8.1%, respectively, compared with the weights of animals in the NC (without phytase), and 0.8, 4.5 and 5.3%, respectively, compared with the weights of animals in the positive control diets (with correct levels of Ca and P and without phytase). This linear effect up to the highest dose of phytase (1000 FTU/kg of feed); Dersjant-Li et al. [12] indicate that levels higher than this limit may confer even greater performances. This condition was confirmed in our study; however, the trend did not extend to the highest dose (4500 FYT/kg of feed). The results showed increases in DWG of 10.8, 10.9 and 2.1% for the NC15, NC30 and NC45 treatments, respectively, compared with that in the NC treatment.

The results of our study are in line with those reported by Silva et al. [8]. These authors found that animals fed diets based on corn and soybean meal with reduced levels of Ca and P by 0.11 and 0.13%, respectively, and supplemented with different levels of phytase derived from *C. braakii* (1000, 2000 and 3000 FYT/kg of feed) showed similar performances to animals fed the PC diet (without reductions in Ca and P), with linear responses (*p* < 0.05) for the DFI and DWG parameters.

Recent advances in phytase production technology, including screening and selection of microorganisms [8,12,14], have favored the isolation and expression of enzymes with high levels of activity in acidic environments, such as pig stomachs. These high levels of expression have the potential to release several nutrients, including minerals, amino acids and energy [10], resulting in better performance values, as observed for the phytase used in this study.

Olsen et al. [31] and Almeida et al. [14] determined that phytase leads to better responses in animals that receive diets limited in nutrients (Ca and P), consistent with the findings of the present study for the treatments with reduced nutrition (NC15 and NC30). However, the results revealed that the response to a higher level of phytase (NC45) did not have a corresponding effect on animal performance (Table 2).

The best use of phosphorus from plant ingredients was reported by Almeida et al. [14]. Lower phosphorus concentrations were found in the faeces of pigs in the growth phase that were fed diets containing phytase than in those of animals fed a negative control diet (2.9 vs. 3.4 g/day), with the best digestibility of phosphorus (69.1%) achieved with 801 FYT/kg of feed. Regarding Ca digestibility (83.5%), the best phytase concentration was 574 FYT/kg of feed.

The performance of pigs fed with phytase in the finishing phase was not different than those of pigs fed diets proposed by Dersjant-Li et al. [12] (250, 500 and 1000 FYT/kg), although the authors reported that pigs fed the NC diet had numerically lower final LWs (kg) than pigs fed the other treatment diets; similar findings were observed in the present study, including significant differences between groups (*p* < 0.05). The DFI and FC parameters were not affected among animals fed diets that contained phytase (Table 2; *p* > 0.05), a result similar to those reported previously [8,12,14].

The appropriate use of exogenous enzymes, such as phytase in feeds, requires strategic reductions in energy and nutrient levels as well as recognition of feed ingredients for the optimization of enzyme benefits, with repercussions on productive and economic indices. However, it is also important to know the potential limitations of enzymes; when improperly used, they can worsen animal performance due to several factors, such as compositional and nutritional misalignment of diets, resulting in economic losses [1]. This result was observed for the NC45 treatment under the conditions in the present study.

The beneficial effects of adding phytase to the diet of pigs, reviewed by Cowieson et al. [27]; Cowieson et al. [32] and Adeola and Cowieson [1] are reported in hundreds of scientific publications. Commonly, the commercial doses of this enzyme vary between 500 and 750 FYT/kg of feed, releasing between 0.05 and 0.15% of P present in the form of phytate [1]. In practically all the experimental phases, the NC45 treatment, representing an “overdose” condition, led to zootechnical results that varied between intermediate and low levels compared with those of the other treatments (Table 2).

The reduction in anti-nutritional effects related to phytate and consequently release of myo-inositol may not necessarily result in better performance unless the concentrations of digestible amino acids and dietary energy are adequate to promote additional growth. Thus, the direct use of more phytase in the diet does not guarantee a beneficial response to the animal. As shown, the application of supra-nutritional phytase concentrations should be combined with formulation strategies and analytical information about dietary phytate to optimize applications for the desired responses [27].

Although the phytate concentration required in a diet to induce an overdose response is not yet completely clear, it is possible that there is a specific limit over which the enzyme responses would not be consistent with the higher levels of addition [27], as observed in the NC45 treatment.

Cowieson et al. [27] mapped the effects of phytase and observed an increase in the release of available P as the enzyme dose increased but determined that there was a release limit when the dietary substrate (phytate) was depleted. It is clear that the addition of phytase increases the concentrations of free myo-inositol in the plasma with consequent improvements in performance [33], a result attributed to the increased insulin sensitivity invoked by the molecule [34]; however, it is not clear how sensitive these responses are to the composition of the diet used (Table 1). This is particularly important when information on the role of dietary P and Ca concentrations (and the interference of the associated phytase) impact the microbiology of the intestine [35].

In the finishing phase, the pigs supplemented with 1500 and 3000 FYT (i.e., the NC15 and NC30 treatments) had higher live weights at 112 days old (*p* < 0.05), as did the pigs in the PC treatment (without reductions in Ca and P). Studies conducted by Holloway et al. [9] showed that phytase levels between 1000 and 2500 FYT/kg of feed improved the performance (DWG and FCR) of finishing pigs, resulting in a higher slaughter weight. The better performance in this period can be credited to the fact that progressively higher amounts of phytase in the diet increased the plasma levels of blood inositol of the animals in the finishing phase [15]. These results support that phytic acid was hydrolyzed in vivo to its final product, allowing a greater nutritional intake and, consequently, a greater weight at 112 days old (LW112). In addition, animals supplemented with 500 and 2000 FYT/kg showed lower concentrations of inositol (InsP6 and InsP5) in the ileum and better performance when fed a diet low in P [19].

Myo-inositol upregulates the expression of insulin and IGF-1 protein genes-related pathways [36], routes that are responsible for increased muscle protein deposition and gluconeogenesis downregulation. 

The use of phytase in the swine diets and the oral administration of purified myo-inositol in nutritionally relevant concentrations demonstrated to increase plasma concentrations of free myo-inositol, increasing insulin sensitivity [34,37], leading to improved pig performance [33,38]. Therefore, the phytase presents two different actions, one acting as a specific enzyme, the other through the release of myo-inositol.

The performance results found in this study were for males only. We choose to remove the sex effect according to the Rosenfelder-Kuon [39] approach, which evaluated the effect of sex by simultaneous computation of regressions, these evaluations led to shapes of estimated regressions that lacked biological meaning. The health status of the animals used in this evaluation was high, no medication was recorded and only two deaths were related to sudden death.

Dersjant-Li et al. [12] and Silva et al. [8] evaluated, respectively, the characteristics of the carcasses of pigs fed with negative control diets supplemented with phytases derived from *Buttiauxella* spp. (250, 500 and 1000 FYT/kg of feed) and *C. braakii* (1000, 2000, and 3000 FYT/kg of feed). Neither study reported negative influences of the enzymes on carcass characteristics; these results were confirmed in our study by the observation of the absence of a regression effect for all the parameters evaluated (Table 3; *p* > 0.05).

Phytase originated and expressed in *A. niger* (1000 FYT/kg of feed) during the growth period did not influence carcass characteristics [29]. However, Lozano et al. [40] supplemented feed with *A. niger*-based phytase for finishing pigs and observed a greater loin depth in treatment groups that received 500 and 1000 FYT/kg of feed than in the NC group. Similarly, Brady et al. [41] reported a linear increase in dorsal fat thickness and a linear decrease in lean meat content with the use of a *Peniophora lycii* phytase (500, 750, and 1000 FYT/kg of feed). These authors attribute these effects to progressive use of dietary energy (myo-inositol) released by the increase in phytase levels.

This finding, however, was not supported in our study, which indicates that issues related to the compositional and nutritional aspects of the diets should be considered for these characteristics, as shown by Cowieson et al. [27].

However, a favorable result was observed for the percentage of lean meat in the carcass for the highest dose of phytase used (the NC45 treatment); more carcasses of pigs in this treatment had yields between 55–60% (*p* < 0.05) (according to the European Carcass Classification System) than those of pigs in the other treatments. This result also emphasizes the need to know the nutritional conditions of the diet but may suggest the role of the higher levels of myo-inositol provided by the enzyme, which improve the deposition of lean meat (by increasing muscle content and reducing fat content), an effect related to the increase in insulin sensitivity that the molecule confers [34]. Additionally, due to lower zootechnical performance, animals had lower final LWs, and the carcasses of the pigs in this treatment followed a well-known relationship, i.e., lighter animals generally deposit or have less body fat body, with positive impacts on the percentage of lean meat.

## 5. Conclusions

Supplementation with phytase between 1500 and 3000 FYT/kg of feed in diets based on corn and soybean meal with a reduction in inorganic phosphorus and calcium improves the DWG and LW of animals in the growing phase, which affect performance on finishing phases and slaughter body weight, representing an economic and environmental benefit. Phytase levels higher than 3000 FYT/kg of feed require further study. 

## Figures and Tables

**Table 1 animals-12-02552-t001:** Composition (inclusion %) of the diets used during the experimental period.

Ingredients (%)	Growing I (68–91 Days)		Growing II (92–112 Days)		Finishing I (113–140 Days)		Finishing II (141–156 Days)	
	PC	NC	PC	NC	PC	NC	PC	NC
Corn	68.90	69.99	74.18	75.29	78.55	78.99	81.86	82.97
Soybean meal	23.99	23.80	19.45	19.25	15.62	15.54	12.63	12.43
Dicalcium phosphate	1.49	0.79	1.26	0.56	1.12	0.42	1.07	0.36
Limestone	0.67	0.83	0.61	0.77	0.57	0.73	0.56	0.72
Soybean oil	3.46	3.10	3.00	2.62	2.65	2.83	2.44	2.07
L-Lysine	0.42	0.42	0.43	0.43	0.43	0.44	0.43	0.43
DL-Methionine	0.08	0.08	0.14	0.14	0.12	0.12	0.12	0.12
L-Threonine	0.15	0.15	0.13	0.13	0.13	0.13	0.11	0.10
L-Tryptophane	0.05	0.05	0.05	0.05	0.05	0.05	0.05	0.05
L-Valine	0.04	0.04	0.04	0.04	0.05	0.05	0.05	0.05
Vitamin Premix ^1^	0.10	0.10	0.10	0.10	0.10	0.10	0.10	0.10
Mineral premix ^2^	0.10	0.10	0.10	0.10	0.10	0.10	0.10	0.10
Salt	0.39	0.39	0.36	0.36	0.34	0.34	0.33	0.33
Adsorbent ^3^	0.15	0.15	0.15	0.15	0.15	0.15	0.15	0.15
Antioxidant-BHT 99% ^4^	0.01	0.01	0.01	0.01	0.01	0.01	0.01	0.01
Calculated composition								
ME (kcal/kg)	3.35	3.35	3.35	3.35	3.35	3.35	3.35	3.35
Dry matter, %	90.33	90.23	90.12	90.02	89.95	89.91	89.83	89.73
Crude protein, %	16.97	16.97	15.34	15.34	13.94	13.94	12.82	12.82
Ethereal extract, %	6.73	6.41	6.34	6.01	6.06	6.25	5.89	5.56
Ash, %	5.21	4.68	4.67	4.14	4.28	3.75	4.05	3.52
Calcium, %	0.73	0.62	0.63	0.52	0.57	0.46	0.55	0.44
Total phosphorus, %	0.57	0.44	0.52	0.39	0.48	0.35	0.46	0.33
Available phosphorus, %	0.36	0.23	0.13	0.18	0.28	0.15	0.27	0.14
Digestible lysine, %	1.06	1.06	0.96	0.96	0.88	0.88	0.81	0.81
Digestible methionine, %	0.38	0.38	0.35	0.35	0.34	0.34	0.30	0.30
Digestible meth + cys, %	0.63	0.63	0.57	0.57	0.53	0.53	0.49	0.49
Digestible threonine, %	0.69	0.69	0.63	0.63	0.66	0.66	0.53	0.53
Digestible tryptophan, %	0.21	0.21	0.19	0.19	0.18	0.18	0.16	0.16
Digestible valine, %	0.74	0.74	0.67	0.67	0.61	0.61	0.56	0.56

ME: Metabolizable energy; ^1^ Vitamin premix provided per kg of diet: 6000 IU vitamin A; 1500 IU vitamin D3; 15 mg vitamin E; 1.5 mg vitamin K3; 1.35 mg vitamin B1; 4 mg vitamin B2; 2 mg vitamin B6; 20 μg vitamin B12; 20 mg; 9.35 mg pantothenic acid; 600 μg folic acid; 80 μg biotin; 300 μg Se. ^2^ Mineral premix provided per kg of diet: 100 mg Fe; 10 mg Cu; 40 g Mn; 1 mg Co; 100 mg Zn; 1.5 mg. ^3^ Bionit. BHT: ^4^ Butylated hydroxytoluene.

**Table 2 animals-12-02552-t002:** Mean live weight (LW), daily weight gain (DWG), daily feed intake (DFI), and feed conversion rate (FCR) during each phase and total evaluation period according to the experimental treatment.

Parameters	Treatments					CV (%)	*p*-Value	*p*-Value (Regression Effect) *	
	PC	NC	1500 FYT	3000 FYT	4500 FYT			Linear	Quadratic
	Growing I (68–91 days)								
LW68 (kg)	25.42	25.00	25.21	25.00	25.21	3.3	0.833	0.954	0.995
DWG (kg)	1.06 ^a^	0.95 ^b^	1.06 ^a^	1.06 ^a^	0.98 ^ab^	7.5	0.012	0.714	0.095
DFI (kg)	1.93	1.79	1.85	1.87	1.77	7.9	0.173	0.897	0.657
FCR	1.83	1.88	1.74	1.76	1.81	6.4	0.159	0.278	0.033 ^1^
	Growing II (92–112 days)								
LW91 (kg)	49.80 ^a^	46.97 ^b^	49.59 ^a^	49.39 ^ab^	47.66 ^ab^	3.9	0.015	0.819	0.491
DWG (kg)	1.09	1.07	1.16	1.14	1.10	6.8	0.158	0.805	0.110
DFI (kg)	2.61	2.54	2.66	2.58	2.25	6.3	0.178	0.477	0.338
FCR	2.39	2.37	2.30	2.28	2.25	6.0	0.218	0.072	0.174
	Finishing I (113–140 days)								
LW112 (kg)	72.75 ^ab^	69.45 ^c^	73.90 ^a^	73.25 ^ab^	70.44 ^bc^	3.7	0.007	0.847	0.282
DWG (kg)	1.15	1.15	1.18	1.17	1.13	7.2	0.810	0.647	0.569
DFI (kg)	3.12	3.11	3.21	3.15	2.93	7.8	0.223	0.301	0.247
FCR	2.72	2.69	2.71	2.70	2.59	7.4	0.678	0.394	0.482
	Finishing II (141–156 days)								
LW140 (kg)	104.93	101.73	106.96	106.01	102.18	4.0	0.062	0.974	0.275
DWG (kg)	1.06 ^ab^	1.13 ^a^	1.20 ^a^	1.14 ^a^	0.94 ^b^	12.4	0.008	0.006 ^6^	0.000 ^2^
DFI (kg)	3.30	3.29	3.48	3.34	3.07	7.8	0.066	0.106	0.023 ^3^
FCR	3.18	2.91	2.89	2.96	3.30	11.4	0.103	0.017 ^7^	0.015 ^4^
W156 (kg)	118.73 ^abc^	116.47 ^bc^	122.95 ^a^	120.79 ^ab^	114.43 ^c^	4.0	0.011	0.584	0.107
	Total (68–156 days)								
DWG (kg)	1.10 ^abc^	1.08 ^bc^	1.15 ^a^	1.13 ^ab^	1.05 ^c^	4.9	0.011	0.501	0.025 ^5^
DFI (kg)	2.70	2.64	2.75	2.69	2.52	6.0	0.075	0.353	0.214
FCR	2.46	2.45	2.39	2.39	2.39	4.7	0.590	0.383	0.558

^a,b,c^ Means with different letters in a row are different by Tukey’s Test (*p* < 0.05). * Linear or quadratic response according the phytase level in the diet. R^2^ = coefficient of regression. PC = positive control. NC = negative control. FYT = Phytase Unit. CV = coefficient of variation. NS = not significative. ^1^ Y = 1871 − 0.000103χ + 0.00000002χ^2^; *p*-value = 0.033; R^2^ = 0.74. Better phytase level = 2575 FYT. ^2^ Y = 1.134 + 0.000088 − 0.00000003χ^2^; *p*-value = 0.0000; R^2^ = 0.48. Better phytase level = 1466 FYT. ^3^ Y = 3.297 + 0.000174 – 0.00000005χ^2^; *p*-value = 0.023; R^2^ = 0.35. Better phytase level = 1740 FYT. ^4^ Y = 2.920 − 0.000098 + 0.00000004 χ^2^; *p*-value = 0.015; R^2^ = 0.28. Better phytase level = 1225 FYT. ^5^ Y = 1.078 + 0.000066χ − 0.0000000164χ^2^; *p*-value = 0.025; R^2^ = 0.60. Better phytase level = 2012 FYT. ^6^ Y = 1.196 − 0.000042χ; *p*-value = 0.006; R^2^ = 0.57. ^7^ Y = 2.834 + 0.000082χ; *p*-value = 0.017; R^2^ = 0.66.

**Table 3 animals-12-02552-t003:** Means of final weight, carcass weight, carcass yield, backfat thickness, loin depth, and lean meat percentage and yield according to the experimental treatment.

Parameters	Treatments					CV (%)	*p*-Value	*p*-Value (Regression Effect) *	
	PC	NC	1500 FYT	3000 FYT	4500 FYT			Linear	Quadratic
Final weight (kg)	120.78 ^a^	116.47 ^ab^	121.30 ^a^	121.21 ^a^	114.86 ^b^	6.27	0.009	0.758	0.027 ^1^
Carcass weight (kg)	86.43 ^ab^	83.48 ^ab^	87.59 ^a^	86.15 ^ab^	82.73 ^b^	6.58	0.022	0.680	0.046 ^2^
Carcass yield (%)	71.53	71.70	72.23	71.09	72.03	2.60	0.283	0.708	0.764
Backfat thickness (mm)	19.76	18.45	17.94	18.31	16.96	22.71	0.321	0.299	0.519
Loin depth (mm)	54.42	56.40	54.80	51.35	53.98	17.67	0.504	0.225	0.284
Lean meat (%)	51.75	52.94	53.17	52.54	53.83	6.47	0.395	0.554	0.636
Lean meat (kg)	45.02	44.13	46.56	45.32	44.54	9.43	0.370	0.954	0.274
Color (L*)	59.75	55.47	57.78	55.26	55.25	13.59	0.233	0.657	0.676

^a,b^ Means with different letters in a row are different by Tukey’s Test (*p* < 0.05). * Linear or quadratic response according the phytase level in the diet. R^2^ = coefficient of regression. PC = positive control. NC = negative control. FYT = Phytase Unit. CV = coefficient of variation. NS = no significative. ^1^ Y = 116.2 + 0.00541χ − 0.00000126χ^2^; *p*-value = 0.027; R^2^ = 0.84. Better phytase level (FYT = 2146). ^2^ Y = 83.66 + 0.00353χ − 0.00000084χ^2^; *p*-value = 0.046; R^2^ = 0.78. Better phytase level (FYT = 2101).

**Table 4 animals-12-02552-t004:** Effect of treatments on the number of carcasses categorized by lean meat yield according to the European Carcass Classification.

Classes *	Treatments				
	PC	NC	1500 FYT	3000 FYT	4500 FYT
S	0	2	0	0	0
E	4 ^b^	4 ^b^	6 ^b^	4 ^b^	11 ^a^
U	7	11	13	15	9
R	4	5	2	1	1
O	1	0	0	1	1
P	0	1	0	0	0

* S, E, U, R, O and P, represent, respectively, the following % of lean meat in the carcass, >60; 55–60; 50–55; 45–50; 40–45 and <40. ^a,b^ Means with different letters in a row are different by Qui-square (*p* < 0.05). PC = positive control. NC = negative control. FYT = Phytase Unit.

## Data Availability

The datasets generated during and/or analyzed during the current study are available from the corresponding author on reasonable request.

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
