# Peer review of "Increasing Doses of Bacterial Phytase (Citrobacter braakii) Improves Performance and Carcass Characteristics of Pigs in Growing and Finishing Phases"

_animals, 2022, doi:10.3390/ani12192552_

Round 1

Reviewer 1 Report

In this manuscript an experiment with growing pigs is described, in which the supplementation of four diets low in Ca and P with a commercial phytase on growth performance and slaughter parameters was tested. The experimental design is clear and correct with a positive and a negative control treatment. Unfortunately, the authors have not invested enough analytical and interpretation activities to make the manuscript interesting. In the present form it remains just a feed trial.

Therefore, in the present form the manuscript cannot be accepted in a scientific journal.

-          English language must be improved. Sentences are often to long or not correct.

-          Do not use values with to many digits after comma in text and tables! (E.g. a body weight 116.470 kg or an air humidity of 64.83 +/- 12.69 % are an absolute nonsense).

-          Sex, health and mortality of animals are not discussed at all.

-          Introduction: There exist many different phytases. Furthermore, beside the phytases other enzymes like carbohydrases can be active. This is not discussed at all. Only the used product name is mentioned.

-          There were no feed analyses made at all and feed treatments was not described (e.g. pelleting). Therefore enzyme activities remain unknown.

-          Tab. 1: add at least analysis for DM, OM, CP, lipids, phytase, ev. other enzymes.
What adsorbent and antioxidant were used? No antibiotic in the diets?

-          Discussion: The missing data in the present experiment do not allow a scientific discussion with the literature available.

-          Conclusion: Not informative at all in the present form.

Author Response

Dear Reviewer

Attached you may find the last version of the paper. Naturally, we have taken into consideration all comments and suggestions. Please give to us, if required, further instructions on how to improve the document.

Reviewer 2 Report

General comments

·         Overall, the use of exogenous phytase sources and super dosing are not new concepts and therefore the experiment does not present any novel findings, especially when the basal diets were corn-soybean, which has been reported extensively.

·         The general power of the study is not sufficient in my opinion, most growing-finishing studies evaluating performance have high variability in feed intake and growth data. What consideration was made to select 120 barrows for this trial? Please show the power analyses done before arriving on 120 animals for this trial.

Specific

L18 delete…and among others

L22-23 The sentence does not sound correct, please rephrase.

L84 which was considered …’the’ not ‘he’ experimental unit

Table 2..why are you using CV instead of standard errors since you analyzed standard ANOVA?

Author Response

(The authors gave the same response as above.)

Round 2

Reviewer 2 Report

Thanks for the response. Nothing further.